# Clinical and Genetic Analysis of a Patient with CMT4J

**Leema Reddy Peddareddygari [1] and Raji P. Grewal [1,2,\*]**

[1] Dynamic Biologics Inc., 1 Deer Park Drive, Monmouth Junction, NJ 08852, USA; lreddy@dynamicbiologics.com
[2] Saint Francis Medical Center, Neuroscience Institute, 601 Hamilton Avenue, Trenton, NJ 08629, USA
\* Correspondence: RGrewal@stfrancismedical.org

**Abstract:** We report the clinical and genetic analysis of a patient with a rare form of an autosomal recessive genetic neuropathy, Charcot Marie Tooth (CMT) disease type 4J. She presented at age 62 years with signs and symptoms consistent with a mild neuropathy. The onset of symptoms began approximately ten years earlier. Electrophysiological testing confirmed a demyelinating neuropathy and a comprehensive neuropathy screening for common causes of neuropathy was unrevealing. She underwent commercial whole exome sequencing, analyzing more than eighty genes known to cause neuropathy. Two mutations were detected, c.122T > C, p.Ile41Thr and c.2247dupC, p.Ser750GlnX10 in the *FIG4* gene. The p.Ile41Thr mutation, which is paternally inherited, is a recurrent mutation reported in a number of unrelated families of European descent. The patient's father, also of European descent, provides further evidence supporting a founder effect for this mutation. In most patients carrying the p.Ile41Thr mutation, the neuropathy, unlike our patient, is often severe with early onset. The second mutation, c.2247dupC, p.Ser750GlnX10 is maternally inherited and not previously reported. Furthermore, based upon our protein modeling analysis, c.2247dupC is disease producing, representing a novel pathogenic mutation. Our study of this patient expands the clinical and genetic spectrum of patients with CMT 4J.

**Keywords:** CMT4J; *FIG4* gene; next generation sequencing analysis; novel variant

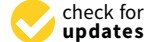



## 1. Introduction

Charcot Marie Tooth disease type 4J (CMT4J) is an uncommon genetic neuropathy caused by mutations in the *FIG4* gene. This gene encodes a phosphatidylinositol protein that is involved in the cycling of intracellular organelles [1]. Interestingly, mutations in this gene are associated with several disorders including amyotrophic lateral sclerosis and Yunis–Varon syndrome in addition to neuropathy [2]. We present a patient who suffered a slowly progressive demyelinating neuropathy caused by mutations in the *FIG4* gene.

## 2. Materials and Methods

### 2.1. Clinical

The index patient, age 62 years, was referred for a second opinion for a possible genetic neuropathy with complaints of gait imbalance. The exact onset of these symptoms is uncertain but a family member indicated that at age 42 years, she had no difficulty walking. Then at around age 52 years she developed a weak right ankle. Her gait slowly worsened and ultimately, she was sent to an orthopedic surgeon who considered the diagnosis of Charcot Marie Tooth (CMT) disease and referred her for neurological evaluation. The review of symptoms revealed no history of numbness, tingling or weakness in her arms. Her general medical history was unremarkable with absence of diabetes mellitus or conditions known to cause a neuropathy.

Physical examination disclosed normal vital signs and evidence of pes cavus. Neurological examination revealed a normal mental status. Tests of cerebellar function and the cranial nerves were also carried out and were normal. Sensory examination was normal

in the hands and showed a mild decrease in vibratory sense with preservation of proprioception and pinprick sensibility in the feet. Her stretch reflexes were reduced at the biceps, brachioradialis, triceps and patellae and not obtained at the ankles. The plantar responses were flexor. Power testing showed normal strength in the arms except for mild equivocal decrease in hand grip bilaterally. In the legs, testing the proximal muscles including hip flexion, extension, abduction and adduction, knee flexion and extension were all Medical Research Council (MRC) Grade 5/5. Testing foot dorsiflexion, eversion and inversion displayed MRC Grade 4/5 weakness bilaterally. She had a positive Romberg test, was unable to stand on her toes or heels and could not perform a tandem walk.

Family history reveals that her father, age 90 years, mother, age 86 years, and a sister, age 63 years have no neurological signs or symptoms of neuropathy (Figure 1). A 64-year-old sister with a history of multiple myeloma developed an acute neurologic event that was initially diagnosed as Guillian–Barre Syndrome and then an acute transverse myelitis. She was evaluated and examined, demonstrating both upper motor neuron and lower motor neuron features and underwent genetic testing. A 32-year-old daughter, the only child, is also asymptomatic and there is no history of any extended members of the family affected by any neurological disorder.

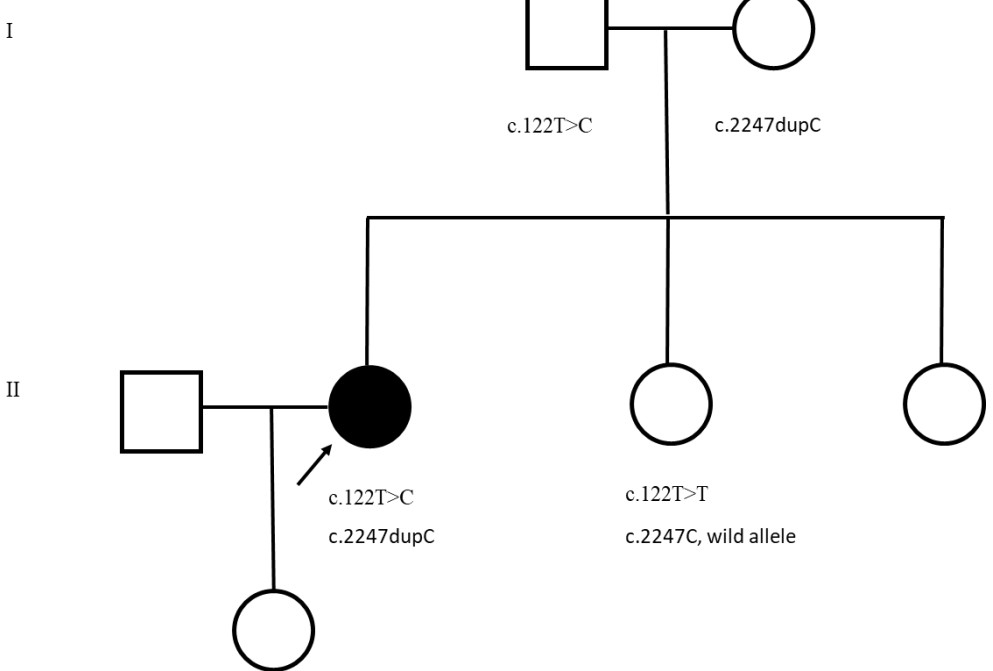

**Figure 1.** This is the pedigree showing parents each carrying a variant in the *FIG4* gene and the affected daughter has inherited both the variants. Squares, males; Circles, females; Dark circle, affected female.

An electromyogram (EMG) was performed (Table 1) and confirmed a sensorimotor neuropathy with demyelinating features including the presence of prolonged F-wave latencies, marked slowing of the conduction velocities and conduction block/temporal dispersion in the left ulnar and median nerves. The abnormalities are more marked in the sensory nerves of the arms compared to the motor nerves. Needle EMG showed the presence of chronic neurogenic changes (high amplitude motor units with a decreased interference pattern with maximal effort) most marked in the distal limb muscles, especially prominent in the legs. Routine serum chemistries and a cell count and differential were normal.

**Table 1.** Nerve conduction studies in the index patient.

| Nerve | Distal Latencies, ms (nl) | Response Amplitude, mv | Conduction Velocity, m/s | F-Wave Latency, ms | Comments |
|---|---|---|---|---|---|
| Motor | | | | | |
| L. Median | 5.1 (<4.2)<br>5.0 | 4.0 (>4.0)<br>2.3 | 31.0 (>50) | 41.0 (<30) | |
| L. Ulnar | 4.3 (<3.3) | 3.6 (>3.5)<br>1.4 (BE)<br>1.0 (AE) | 30.0 (>50)<br>21 | 37.58 (<30)<br>41.4 | |
| Bilateral Fibular | NR (<6.2) | NR (>2.6) | NR (>40) | NR | No response recording the extensor digitorum brevis |
| Bilateral Tibial | NR (<6.0) | NR (>4.0) | NR (>40) | NR | |
| Sensory | | | | | |
| Bilateral Sural | NR | | | | |
| Bilateral Fibular | NR | | | | |
| L. Median | 4.8 | 10.2 (>20) | 38.0 (>50) | | |
| L. Ulnar | 3.1 | 2.7 (>17) | 50.0 (<50) | | |
| L. Radial | 3.9 | 5.0 (>15) | 44.0 (>50) | | |

nl—normal latency, NR—no response; all sensory response latencies are onset latencies, AE—above elbow, BE—below elbow.

### 2.2. Genetic Analysis

Genetic analysis was performed with commercially-available next generation sequencing analysis, screening twenty-four genes known to cause neuropathy and was negative. This panel did not include testing for the *FIG4* gene. Then, using a different company screening more than 80 genes known to cause neuropathy resulted in the identification of two mutations in the *FIG4* gene. Further analysis was done to confirm that these mutations are in trans by analyzing her parents. Informed consent was obtained from all subjects involved in the study. The study was conducted according to the guidelines of the Declaration of Helsinki, and approved by the Institutional Review Board of JFK Medical Center (protocol code FWA00001350 and date of approval, 15 January 2007).

### 3. Results

The first mutation, c.122T > C, p.Ile41Thr, is paternally inherited while a second mutation c.2247dupC, p.Ser750GlnX10, is maternally inherited. The p.Ile41Thr (rs121908287) has been previously published as a disease-producing mutation and functional studies show protein instability [3]. The maternal mutation is novel and results from duplication (indicated in brackets) in the following normal sequence: GCCCCC[dupC]AGCG. This variant c.2247dupC (c.2247_2248insC) was analyzed using the Mutation Taster analysis tool. A number of consequences are predicted to occur including a frameshift mutation that could result in a nonsense-mediated mRNA decay or a premature termination codon, S750Qfs*10, ultimately resulting in premature truncation of the protein. The analysis also predicts that splice site changes could occur since the mutation is present 68 bp upstream from the splice site [4]. This is a novel variant not reported in the ExAC or the 1000G databases. The patient's sister with neurological abnormalities was genotyped and did not carry either of the mutations under discussion (Figure 1).

### 4. Discussion

We confirm that in this family, mutations in the *FIG4* gene cause CMT4J, a rare form of genetic neuropathy. In a study of 4000 patients with CMT, 8 patients were diagnosed with CMT4J carrying compound heterozygote mutations in the *FIG4* gene representing a frequency of 0.2% [3]. In a more recent study of 17,880 individuals with genetic neuropathy, CMT4J was found in 0.3% of those genotyped [5]. Our study of the index patient confirms previous reports of a neuropathy with demyelinating features that would suggest an acquired neuropathy such as chronic inflammatory demyelinating polyneuropathy with the presence of conduction block and temporal dispersion in the nerve conduction studies [3,6].

Her clinical course is on the milder end of the spectrum of patients which ranges from mild disability to more significant neurological deficits requiring the use of a wheelchair [3]. Some patients with CMT4J can exhibit features of amyotrophic lateral sclerosis prompting us to consider whether her sister affected with the neuropathy/transverse myelitis could have also carried mutations in the *FIG4* gene. Interestingly, she has neither of the mutations, eliminating this possibility as a cause of her neurological condition.

Our patient carries the previously described paternally-inherited p.Ile41Thr mutation. In the report by Nicholson et al., 15 of 16 CMT4J patients carried this mutation with a calculated frequency of 0.001 reported in people of Northern European descent with haplotype analysis indicating a founder effect in this population. Interestingly, a recent report based in Greece suggests that this mutation is not exclusive to Northern Europeans [7]. This variant has been reported in both the 1000G (C = 0.0008) and ExAC (C = 0.000991) databases. The ALFA project that provides allele frequency from dbGaP shows that although this variant is reported with higher frequency in the European population C = 0.001896 ($n$ = 149,800), it has also been detected in other populations. In Africans ($n$ = 4862) and African Americans ($n$ = 4686) it occurs with a frequency C = 0.0002. It has also been found in two separate Latin American populations ($n$ = 946) with C = 0.002 and others ($n$ = 11,922) C = 0.00134 [8]. In this family, the patient's father is of Northern European descent and likely carries this ancestral mutation originating in Europe. It has been reported that compound heterozygotes with this mutation diagnosed with CMT4J usually show early onset with rapid progression [9]. In contrast, our patient demonstrates that even in those carrying this mutation, the onset can be late and progression slow.

## 5. Conclusions

Our study demonstrates the power of next generation sequencing to identify the genetic basis of patients with CMT even when the frequency is rare. We also expand the spectrum of mutations that can cause CMT4J and the clinical phenotype associated with these mutations.

**Author Contributions:** R.P.G. was involved in the clinical management of the patients, manuscript writing, review and editing. L.R.P. was involved in genetic analysis, variant validation, manuscript review and editing. All authors have read and agreed to the published version of the manuscript.

**Funding:** This research was funded by Neurogenetics Foundation.

**Institutional Review Board Statement:** The study was conducted according to the guidelines of the Declaration of Helsinki, and approved by the Institutional Review Board of JFK Medical Center (protocol code FWA00001350 and date of approval, 15 January 2007).

**Informed Consent Statement:** Informed consent was obtained from all subjects involved in the study.

**Acknowledgments:** We are grateful to all participating members of the family for their cooperation in this study.

**Conflicts of Interest:** The authors declare no conflict of interest.

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
