# Peer review of "Clinical and Genetic Analysis of a Patient with CMT4J"

_2035-8377, doi:10.3390/neurolint14010017_

Round 1

Reviewer 1 Report

The authors present an unusual example of late onset autosomal recessive CMT. The detailed description of the mutations is clear and interesting. A few questions and clarifications are raised.

Fibular should be used instead of peroneal.

Did the original panel that missed these mutations test Fig4, simply miss the finding, or consider the mutation to be an unreportable VUS? 

It appears that the patient was thought to have genetic neuropathy prior to the second opinion and confirmatory diagnosis. We she erroneously thought to have immune neuropathy based on the demyelinating characteristics at some point?

Author Response

We are grateful for the responses of the reviewer 1.  Our reponse for the comments of reviewer are as follows:

“Fibular should be used instead of peroneal”

We have changed peroneal to “fibular” in Table 1 on page 3.

“Did the original panel that missed these mutations test Fig4, simply miss the finding, or consider the mutation to be an unreportable VUS.”

We have indicated---

“This panel did not include testing for the FIG4 gene.”  on page 3, in Genetic analysis (section 2.2) line 3

Thank you

Reviewer 2 Report

Dear editor, thank you for inviting me to review the manuscript on FIG4 gene variants in an individual with an adult-onset neuropathy CMT4J. The report is sound, though the only apparent novelty is one of the two variants.

Author Response

We are grateful for the comments of the reviewer 2. 

“The report is sound, though the only apparent novelty is one of the two variants.”

Our response to this statement is that this is a rare condition and reporting a novel variant which is important for future patients for diagnosis, genotype phenotype analysis and genetic counseling.

Thank you